# Weekend Effect and in-Hospital Mortality in Elderly Patients with Acute Kidney Injury: A Retrospective Analysis of a National Hospital Database in Italy

**DOI:** 10.3390/jcm9061815

**Published:** 2020-06-11

**Authors:** Fabio Fabbian, Alfredo De Giorgi, Emanuele Di Simone, Rosaria Cappadona, Nicola Lamberti, Fabio Manfredini, Benedetta Boari, Alda Storari, Roberto Manfredini

**Affiliations:** 1Clinica Medica Unit, Azienda Ospedaliero-Universitaria, I-44121 Ferrara, Italy; degiorgialfredo@libero.it (A.D.G.); emanuele.disimone@uniroma1.it (E.D.S.); benedetta.boari@unife.it (B.B.); roberto.manfredini@unife.it (R.M.); 2Department of Medical Sciences, University of Ferrara, I-44121 Ferrara, Italy; rosaria.cappadona@unife.it; 3Department of Biomedical and Specialty Surgical Sciences, University of Ferrara, I-44121 Ferrara, Italy; nicola.lamberti@unife.it (N.L.); fabio.manfredini@unife.it (F.M.); 4Neuroscience and Rehabilitation Department, Azienda Ospedaliero-Universitaria, I-44121 Ferrara, Italy; 5Nephrology and Dialysis Unit, Azienda Ospedaliero-Universitaria, I-44121 Ferrara, Italy; a.storari@ospfe.it

**Keywords:** acute kidney injury, weekend effect, in-hospital mortality, comorbidity, dialysis, elderly

## Abstract

Background: The aim of this study was to relate the weekend (WE) effect and acute kidney injury (AKI) in elderly patients by using the Italian National Hospital Database (NHD). Methods: Hospitalizations with AKI of subjects aged ≥ 65 years from 2000–2015 who were identified by the ICD-9-CM were included. Admissions from Friday to Sunday were considered as WE, while all the other days were weekdays (WD). In-hospital mortality (IHM) was our outcome, and the comorbidity burden was calculated by the modified Elixhauser Index (mEI), based on ICD-9-CM codes. Results: 760,664 hospitalizations were analyzed. Mean age was 80.5 ± 7.8 years and 52.2% were males. Of the studied patients, 9% underwent dialysis treatment, 24.3% were admitted during WE, and IHM was 27.7%. Deceased patients were more frequently comorbid males, with higher age, treated with dialysis more frequently, and had higher admission during WE. WE hospitalizations were more frequent in males, and in older patients with higher mEI. IHM was independently associated with dialysis-dependent AKI (OR 2.711; 95%CI 2.667–2.755, *p* < 0.001), WE admission (OR 1.113; 95%CI 1.100–1.126, *p* < 0.001), and mEI (OR 1.056; 95% CI 1.055–1.057, *p* < 0.001). Discussion: Italian elderly patients admitted during WE with AKI are exposed to a higher risk of IHM, especially if they need dialysis treatment and have high comorbidity burden.

## 1. Introduction

The negative clinical impact of the so-called weekend (WE) effect has been a matter of debate since the past two decades. Different research groups have reported poorer outcomes for patients admitted on WE compared to weekdays (WD). A milestone study published in 2001, conducted on almost 4 million acute care admissions from emergency departments in Ontario, Canada, found that patients with some serious medical conditions had higher in-hospital mortality (IHM) if they were admitted on a WE than on a WD [1]. A few years later, Cram et al. confirmed a modest increase in mortality after WE admission for all admissions, either unscheduled or emergency department admissions [2]. Our group also documented a higher IHM for some cardiovascular events, such as acute heart failure (OR 1.33) [3], and acute pulmonary embolism (OR 1.18) [4]. A systematic review evaluated 97 studies enrolling more than 51 million subjects, and patients admitted on WE had a significantly higher overall mortality, independent of factors including the levels of staffing, procedure rates and delays, and illness severity [5]. Another systematic review and meta-analysis focused on United Kingdom (UK) hospitals confirmed that WE admissions had higher odds of mortality than those admitted during WD, as well as when measures of case mix severity were included in the models. On the other hand, the WE effect was not significant when clinical registry data was used [6]. Finally, Chen et al. performed a large meta-analysis (68 studies, 640 million admissions), and found that risk of mortality during all WE admissions was 1.16, although it was greater for elective admissions than emergency ones [7]. A first consideration is that differences in hospital care associated with the day of the week (measured by indicators including short term mortality) can vary depending on the place, time, and reason for hospital admission [8]. On one hand, medical and nursing understaffing, shortage of diagnostic or procedural services, and the presence of inexperienced residents have been suggested as possible causes [9]. On the other, temporal aspects of onset of acute vascular diseases may also play a role, and it is possible that these diseases do not present with equal severity relative to time, that is, day of the week or hour of the day [10,11]. A single-center study on acute coronary syndromes (ACSs) showed that although there were fewer ACS admissions than expected on nights and WE, the proportion of patients with ACS presenting with ST-elevation myocardial infarctions was 64% higher on WE [12]. Again, in their large study on pulmonary embolism admissions, Nanchal et al. reported a 19% increase in patients admitted on WE [13], but WE admissions showed significantly worse parameters of severity, such as the need for mechanical ventilation, thrombolytic therapy use, and the use of vasopressors. A further confirmation to this hypothesis comes from the results of a study conducted on more than 500,000 unselected emergency admissions in the UK, evaluating and adjusting for multiple confounders including demographics, comorbidities, and admission characteristics, and common hematology and biochemistry test results. Hospital workload was not associated with mortality, suggesting that the WE effect could be associated with patient-level differences at admission rather than reduced hospital staffing or services [14]. Therefore, the debate about clinical impact of the WE effect is still open. Acute kidney injury (AKI) is a frequent finding in hospitalized subjects, especially in people who are 65 years old or older [15,16,17]. However, available data about admissions due to renal diseases are scarce; therefore, we wanted to explore the possible relationship between the WE effect and AK by using the National Hospital Database (NHD).

## 2. Experimental Section

### 2.1. Patient Selection and Eligibility

This retrospective study was conducted in agreement with the Declaration of Helsinki of 1975, revised in 2013. Subject identifiers were deleted before data analysis with the aim of maintaining data anonymity and confidentiality; therefore, none of the patients could be identified, either in this paper or in the database. The study was conducted in agreement with the existent Italian disposition-by-law (G.U. n.76, 31/03/2008), and due to the study design, ethics committee approval was not necessary.

We accessed the National Hospital Database (NHD), provided by the Italian Ministry of Health (SDO Database, Ministry of Health, General Directorate for Health Planning), selecting all hospitalizations complicated by AKI between 1 January 2000, and 31 December 2015. This database stores data of all hospitalizations both in public and private Italian hospitals. Based on the International Classification of Diseases, 9th Revision, Clinical Modification (ICD-9-CM), the hospital discharge record files contain information such as gender, age, date and department of admission and discharge, vital status at discharge (in-hospital death vs. discharged alive), main diagnosis, up to five co-morbidities, and up to six procedures/interventions. For this analysis, patients’ names and all other potential identifiers were removed by the Ministry of Health from the database, following the national disposition-by-law in terms of privacy. A consecutive number for each patient was the only identifier. Although in clinical settings the term AKI has replaced the term acute renal failure, in administrative database codes the latter term is usually the reference term. We selected patients aged ≥65 years in whom the ICD-9-CM code 584.xx identified AKI when used as a first or second discharge diagnosis. As for a temporal definition, midnight Friday to midnight Sunday was considered as WE, while all the other days were assumed as WD. The nine main national festive days in Italy (1 January, 25 April, 1 May, 2 June, 15 August, 2 November, 8 December, 25 December, and 26 December), when occurring on WD, were considered as WE.

### 2.2. Data Analysis

In-hospital mortality (IHM) was the hard clinical outcome indicator. In order to evaluate the comorbidity burden, a novel score from our group, a modified Elixhauser Index (mEI) [18], was calculated based on the guidelines set by Quan et al [19]. To calculate the score, the following conditions were considered: age, gender, presence of chronic kidney disease (CKD), neurological disorders, lymphoma, solid tumor with metastasis, ischemic heart disease, congestive heart disease, coagulopathy, fluid and electrolyte disorders, liver disease, weight loss, and metastatic cancer. The original score was corrected, removing the diagnosis of previous AKI; therefore, the points assigned to renal diseases were considered only if CKD was recorded. The points for each condition ranged from 0 to 16, and the total score calculated could vary between 0 and 89. When the score was >40, the risk of IHM was >60%. The score, based on administrative data, was calculated automatically. Table 1 reports single items and relative score. Finally, dialysis treatment was also taken into consideration (code ICD-9-CM 39.95).

### 2.3. Statistical Analysis

A descriptive analysis of the whole population, i.e., absolute numbers, percentages, and means ± SD, was performed. Univariate analysis was carried out by using the Chi-Squared test, Student *t*-tests, Mann–Whitney U-test, and ANOVA as appropriate, comparing survivors and deceased subjects, and AKI patients admitted during the WE or WD. Moreover, in order to evaluate the relationship between the WE effect and IHM, the latter was considered as the dependent variable in a logistic regression analysis, while demography, comorbidity score, and dialysis-dependent AKI were considered as independent variables. Odds ratios (ORs) with their 95% confidence intervals (95% CI) were reported. All *p*-values were 2-tailed, and *p*-value <0.5 was considered significant. SPSS 13.0 for Windows (SPSS IN., Chicago, IL, USA, 2004) was used for statistical analysis.

## 3. Results

The total sample consisted of 760,664 hospitalizations due to AKI, 52.2% were men, with a mean age of 80.5 ± 7.8 years, and 9% underwent dialysis treatment. Of these patients, 24.3% were admitted during WE and 27.7% died during hospitalization (Table 2).

IHM was significantly higher in men (51.8% vs. 48.2%, *p* < 0.001). Deceased subjects were more likely to be older (81.9 ± 7.9 vs. 80 ± 7.7 years, *p* < 0.001), to have higher comorbidity score (15.96 ± 6.48 vs. 14.04 ± 6.02, *p* < 0.001), to be treated with dialysis (17.7% vs. 6.8%, *p* < 0.001), and to show higher admission during WE (25.8% vs. 23.7%, *p* < 0.001), compared to survivors (Table 3).

Patients admitted during WE were more likely to be male (51.5% vs. 48.5%, *p* < 0.001), older (mean age 81 ± 7.8 vs. 80.4 ± 7.8 years, *p* < 0.001), and had a higher comorbidity score (14.75 ± 6.2 vs. 14.52 ± 6.22, *p* < 0.001). No difference was found for prevalence of dialysis dependent AKI (Table 4).

At the logistic regression analysis, IHM was independently associated, in decreasing order, with dialysis-dependent AKI, WE admission, and comorbidity score (Table 5). As for the comorbidity score, the risk of death raised of 5.6% for every 1-point increase.

## 4. Discussion

In this study, based on a large national database of hospitalizations, the day of admission had a significant clinical impact on elderly subjects with AKI. The WE effect was independently associated with IHM, along with dialysis treatment and comorbidity burden. The OR for IHM was 1.113, and this finding confirmed previous results from our group, also drawn by analysis of the NHD records, regarding pulmonary embolism (OR 1.15) [20], and acute aortic dissection or rupture (OR 1.34) [21]. The importance of a diverse level of emergency has been underlined by previous studies. Concha et al. studied the 7-day post-admission time patterns of excess mortality following WE admission to investigate whether the phenomenon could be due to poorer quality of care or a case selection. After evaluation of mortality risk for WE and WD, adjusting for age, sex, Charlson Comorbidity Index (CCI), and diagnostic group, they found that WE mortality was diverse for different diagnostic groups, and concluded that the WE effect is probably not a uniform phenomenon, but rather a complex cluster of different causal pathways, even associated with quite different risk profiles [22]. Similar results were reported by Roberts et al., who evaluated 30-day mortality for WE admissions in England and Wales. The WE effect was more evident for disorders with high mortality during the acute phase, and negligible for less acute ones [23]. 

Moreover, the presence of comorbidities plays a primary role in determining IHM. In the present study, in fact, the OR of comorbidity score is lower than that of presence of dialysis and age, but the risk of death raised of 4.2% for every 1-point increase. In a previous study conducted by our group, we showed that CCI was significantly higher in subjects admitted during WE, and significantly contributed to clinical outcome, along with gender and age. In logistic regression analysis, in fact, admission on WE, CCI, male sex, and age were significantly associated with IHM [24].

To the best of our knowledge, this is the first study considering the relationship between the WE effect and IHM in elderly patients hospitalized because of AKI. The question of whether the WE effect also exists in renal diseases is still matter of debate, because the number of available studies is limited, and results are not univocal. We are aware of only three studies considered the relationship between WE admission and AKI, conducted in the United States (USA), United Kingdom (UK), and Wales, respectively. James et al. analyzed data from the U.S. Nationwide Inpatient Sample and selected more than 200,000 admissions reporting AKI as the primary diagnosis. The prevalence of WE admission was 21% and WE hospitalizations were independently associated with IHM [25]. Kolhe et al. conducted a study on more than 53,000 dialysis-dependent AKI patients. The prevalence of WE admission was 23%, and WE admissions were significantly associated with higher mortality in the unadjusted model, but not in the multivariable analysis [26]. Finally, Holmes et al. did not find any WE effect for mortality associated with hospital-acquired AKI [27]. None of these studies, however, included comorbidity analysis.

A higher interest in the WE effect has been shown to investigate possible negative outcomes in renal transplantation, but results have not demonstrated any negative outcome thus far. In the U.S., Baid-Agrawal et al. did not confirm the hypothesis that kidney transplants performed during WE could have worse outcomes than those performed during WD. In fact, the day of surgery did not affect death, length of hospitalization after transplantation, delayed allograft function, acute rejection within the first year of transplant, and patient and allograft survival at 1 month and at 1 year after transplantation [28]. In Germany, Schütte-Nütgen et al. found no differences between subjects transplanted on WD or WE in terms of 3-year patient and graft survival, frequencies of delayed graft function, acute rejections, 1-year estimated glomerular filtration rate, and length of hospital stay [29]. Again, in England, Anderson et al. did not confirm the relationship between WE and mortality, rehospitalization, and kidney allograft failure/rejection [30]. Moreover, a study of the Australia and New Zealand Dialysis and Transplant registry concluded that timing of transplantation did not impact on allograft outcome [31]. Also, our group tested this hypothesis on all cases of the Emilia-Romagna region, but did not find any risk of adverse outcome related to the WE effect, observing only that WE admissions were characterized by longer duration of hospitalization [32].

On the other hand, WE admission seems to negatively influence outcomes in dialysis patients, although the available evidence is strictly limited to a couple of studies. In the U.S., Sakhuja et al. reported that WE admissions were more likely to have higher IHM, higher mortality during the first 3 days of admission, longer hospital stays, and less likely to be discharged to home. Moreover, time to death was shorter compared with WD admissions [33]. Finally, data from the Australia and New Zealand Dialysis and Transplant Registry reported higher rates of hospitalization secondary to peritonitis on WE compared to WD [34].

In our present study, dialysis-dependent AKI and the WE effect were independently associated with IHM; it could be that the two factors negatively impact patients’ survival and complications.

### Limitations

We are aware that a major limitation is introduced by the study design: retrospective, based on administrative data, and with no possibility to assess whether AKI was cause or complication of hospitalization. We observed that day-of-week of hospital admission has a significant impact on outcome, but we cannot extrapolate from the administrative database some important items, such as cause of admission and death, intensive care level or hospitals’ facilities, device use, type of treatment, and impact of clinical or biochemical parameters. It is known that administrative databases, born to be used for other reasons (i.e., reimbursement), lack specific clinical information and may cause possible misclassification of outcomes, thereby generating confounding factors [35]. Moreover, we did not identify AKI on the basis of international Kidney Disease Improving Global Outcomes (KDIGO) guidelines [36], nor differentiate patients on the basis of the cause of AKI and the treatment setting, with the exception of dialysis treatment.

We previously stated that medical and nursing understaffing, shortage of diagnostic or procedural services, and the presence of inexperienced residents could be related to WE effect [9]. Unfortunately, administrative databases do not allow us investigate these aspects, being conceived for financial reasons.

Some years ago, concerns about WE effect were raised due to three main potential limitations of administrative databases: (1) coding mistakes, (2) insufficient consideration of comorbidity, and (3) failure to consider the severity or acuity of patients [37]. According to several authors, the performance of ICD-9-CM for diagnosis of acute renal failure showed poor sensitivity, and high specificity, while positive and negative predictive value could differ [38,39,40,41]. However, Grams et al. underlined that sensitivity was significantly higher when the selected individuals were aged ≥ 65 years; moreover, AKI diagnosed by administrative data detected more severe disease and higher IHM mortality [41]. Due to this reason, we decided to focus on patients aged > 65 years. Finally, we also have to underline some strengths of our study: (1) a high number of records derived from a national database, (2) the long period of time analyzed, and (3) the utilization of a hard outcome indicator, such as IHM.

## 5. Conclusions

The global population is ageing, and the prevalence of elderly subjects is increasing. Older adults are projected to increase enormously by 2050, rising to more than 400 million [42]. Chronic illnesses and disability causing hospitalization are frequent in the last decades of life and AKI is a frequent cause of morbidity and mortality as shown by the U.S. Renal Data Services (USRDS) 2018 [43]. The latter data demonstrated an increasing incidence rate of AKI over the past several years, in the elderly population. Patients over the age of 65 who required dialysis continued to have substantially higher mortality compared to general population [43]. Last year, our group demonstrated that in-hospital mortality was a frequent complication in elderly subjects with AKI discharge codes, involving more than a quarter of admissions. The increasing burden of comorbidity, dialysis-dependent AKI, and sepsis were the major risk factors for mortality [16]. Comorbidity is a well-known risk factor affecting survival in dialysis patients [44]; however, predictors of short-term survival in renal patients are still a matter of debate.

Multi-morbidity is crucial for defining the prognosis of the aged population [45], and our findings suggest that pre-existing diseases diagnosed prior to admission may be associated with the outcome of an acute condition such as AKI (especially if AKI needs dialysis treatment). In elderly hospitalized subjects with AKI, WE effect seems to be a risk factor for IHM, even adjusting for comorbidity and advanced AKI stage. Thus, elderly patients admitted on Saturday or Sunday should deserve careful attention and evaluation, and consideration should be taken of their higher risk of IHM.

## Figures and Tables

**Table 1 jcm-09-01815-t001:** Items and relative assigned score to calculate in-hospital mortality (IHM).

Items	Score
Age 0–60 (years)	0
Age 61–70 (years)	3
Age 71–80 (years)	7
Age 81–90 (years)	11
Age 91+ (years)	16
Chronic kidney disease	1
Male gender	2
Neurological disorders	3
Lymphoma	4
Solid tumor without metastasis	4
Ischemic heart disease	5
Congestive heart failure	5
Coagulopathy	8
Fluid and electrolyte disorders	8
Liver disease	10
Cachexia	11
Metastatic cancer	12

**Table 2 jcm-09-01815-t002:** Demographic features of the considered sample (AKI: acute kidney injury, WE: weekend effect).

Total Number of Records	760,664
Men (*n* (%))	397,174 (52.2)
Women (*n* (%))	361,490 (47.8)
Age (years)	80.5 ± 7.8
Comorbidity score	14.57 ± 6.21
Dialysis dependent AKI (*n* (%))	68,563 (9)
Patients admitted during WE (*n* (%))	184,727 (24.3)
Deceased subjects (*n* (%))	210,661 (27.7)

**Table 3 jcm-09-01815-t003:** Comparison between survivors and deceased subjects.

	Survivors *n* = 550,003	Deceased *n* = 210,661	*p*
Men (*n* (%))	288,120 (52.4)	109,054 (51.8)	<0.001
Women (*n* (%))	261,883 (47.6)	101,607 (48.2)	
Age (years)	80 ± 7.7	81.9 ± 7.9	<0.001
Comorbidity score	14.04 ± 6.02	15.96 ± 6.48	<0.001
Dialysis dependent AKI (*n* (%))	37,598 (6.8)	31,055 (17.7)	<0.001
Patients admitted during WE (*n* (%))	130,318 (23.7)	54,409 (25.8)	<0.001

**Table 4 jcm-09-01815-t004:** Comparison between subjects admitted during weekdays (WD) or weekends (WE).

	WD Admissions *n* = 575,937	WE Admissions *n* = 184,727	*p*
Men (*n* (%))	302,010 (52.4)	95,164 (51.5)	<0.001
Women (*n* (%))	273,927 (47.6)	89,563 (48.5)	
Age (years)	80.4 ± 7.8	81 ± 7.8	<0.001
Dialysis dependent AKI (*n* (%))	52,075 (9)	16,578 (9)	NS
Comorbidity score	14.52 ± 6.22	14.75 ± 6.2	<0.001

NS: non-significant.

**Table 5 jcm-09-01815-t005:** Logistic regression analysis showing factors independently associated with in-hospital mortality. OR: Odds Ratio; 95%CI: 95% confidence intervals; WE: weekend.

	OR	95% Confidence Intervals	*p*
Dialysis dependent AKI	2.711	2.667–2.755	<0.001
WE admission	1.113	1.100–1.126	<0.001
Comorbidity score	1.056	1.055–1.057	<0.001

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
