# Peer review of "Weekend Effect and in-Hospital Mortality in Elderly Patients with Acute Kidney Injury: A Retrospective Analysis of a National Hospital Database in Italy"

_jcm, 2020, doi:10.3390/jcm9061815_

Round 1
Reviewer 1 Report
The study question is relevant and its designed well.
The manuscript is overall well prepared and presented.
Author Response
Reviewer 1
The study question is relevant and its designed well. The manuscript is overall well prepared and presented.
Thank you for kind words of appreciation.
Reviewer 2 Report
The authors investigated the association between weekend admission and poorer outcome among 760,664 elderly AKI patients. Although the results of this study were meaningful, there are several problems.
The authors should mention the reason why they investigate the association between weekend admission and patient outcome among AKI patients. It remains uncertain whether these findings will help improving outcome of the AKI patients. Is there any specific therapeutic approach for this population?
Why did the authors target only elderly patients?
As the authors mentioned in the Introduction, medical and nursing understaffing, shortage of diagnostic or procedural services, and presence of inexperienced residents have been suggested as possible causes of poorer outcome among this population. Unfortunately, they did not analyze these effects in analysis.
Author Response
Reviewer 2
The authors investigated the association between weekend admission and poorer outcome among 760,664 elderly AKI patients. Although the results of this study were meaningful, there are several problems.
The authors should mention the reason why they investigate the association between weekend admission and patient outcome among AKI patients. It remains uncertain whether these findings will help improving outcome of the AKI patients. Is there any specific therapeutic approach for this population?
Thank you for your observation. Weekend effects is a meaningful and interesting topic in medical community because it appears to be related to negative outcomes in several common conditions leading to hospitalization. For this reason, it could require action aiming at optimizing clinical organization involved in the care for acute patients, especially in low intensity wards. Moreover, weekend effect in renal patients is scarcely evaluated, therefore, we would like to look for an association between negative outcome and admissions during weekend in hospitalized patients with acute kidney injury in non-selected wards.
Why did the authors target only elderly patients?
Thank you for your question. A new paragraph has been added to the ‘Conclusions’ section, accordingly.
As the authors mentioned in the Introduction, medical and nursing understaffing, shortage of diagnostic or procedural services, and presence of inexperienced residents have been suggested as possible causes of poorer outcome among this population. Unfortunately, they did not analyze these effects in analysis.
Thank you for your question. A new paragraph has been added to the ‘Limitations’ section, accordingly.
Reviewer 3 Report
The manuscript by Fabbian et al investigated the relationship between weekend (WE) effect and acute kidney injury (AKI) , by using the National Hospital Database. The manuscript is well written. The results are convincible In general these studies provide new information.
Author Response
Reviewer 3
The manuscript by Fabbian et al investigated the relationship between weekend (WE) effect and acute kidney injury (AKI) , by using the National Hospital Database. The manuscript is well written. The results are convincible In general these studies provide new information.
Thank you for kind words of appreciation.
Reviewer 4 Report
Thank you for giving me the opportunity to review the manuscript by Fabbian et colleagues entitled: “Week-end effect & mortality in elderly patients with AKI: a retrospective analysis of national hospital database in Italy”.
The authors propose to explore the relationship between week-end effect end AKI from an Italian National hospital database.
This is an original subject, and a large number of records are analyzed. The manuscript is well written, results are clearly described and the authors stick to the objective of their study.
I have few comments:
- The authors state that hospitalizations were selected if they were due to AKI (page 2, line 81) meaning that the hospitalizations were selected only if code 584.xx was the main or second discharge code. As fairly stated in the discussion, this could probably be a source of classification bias, as the initial cause of hospitalization is often a clinical symptom leading to the discovery of AKI after blood test have been performed. Could the authors evaluate how many hospitalizations with a 584.xx code were not selected if they have access to this data?
- Presentation of the data is clear. I appreciate table 3 and 4.
- The dataset is so important that nearly all comparisons performed are statistically significant with the cut-off of p-value to 0.05. For the results of the logistic regression, the OR for comorbidity score is statistically significant, but I am not sure of its clinical significance. The conclusion could be tempered based on this observation.
- The authors chose to use a comorbidity score as adjusting variable. Did they have access to the details of the values used for the calculation of the score. If so, it could have been interesting to perform another logistic regression model with all of these variables, in a sensitivity analysis for example.
Minor comments:
- typo on page 5, line 146: “The OR for IHM was 1.11” instead of “1.13”, according to table 5.
- Page 5, line 53. I am not sure that the abbreviation CCI has been defined in the text (I guess it stands for Charlson Comorbidity Index?)
- typo on page 5, line 174: “conducted study on more than cases in 53,000 dialysis…”.
Author Response
Reviewer 4
Thank you for giving me the opportunity to review the manuscript by Fabbian et colleagues entitled: “Week-end effect & mortality in elderly patients with AKI: a retrospective analysis of national hospital database in Italy”.
The authors propose to explore the relationship between week-end effect end AKI from an Italian National hospital database.
This is an original subject, and a large number of records are analyzed. The manuscript is well written, results are clearly described and the authors stick to the objective of their study.
Thank you for kind words of appreciation.
I have few comments:
- The authors state that hospitalizations were selected if they were due to AKI (page 2, line 81) meaning that the hospitalizations were selected only if code 584.xx was the main or second discharge code. As fairly stated in the discussion, this could probably be a source of classification bias, as the initial cause of hospitalization is often a clinical symptom leading to the discovery of AKI after blood test have been performed. Could the authors evaluate how many hospitalizations with a 584.xx code were not selected if they have access to this data?
The reviewer’s comment is quite right, it would be interesting to evaluate the reason for admission in subjects discharged with diagnosis of AKI, however administrative databases record only discharge diagnoses, it is not possible to analyse causes for admissions. Any word of the text suggesting such an idea has been corrected through the text.
- Presentation of the data is clear. I appreciate table 3 and 4.
Thank you for appreciation.
- The dataset is so important that nearly all comparisons performed are statistically significant with the cut-off of p-value to 0.05. For the results of the logistic regression, the OR for comorbidity score is statistically significant, but I am not sure of its clinical significance. The conclusion could be tempered based on this observation.
Thank you for your comment. The text has been changed, accordingly.
- The authors chose to use a comorbidity score as adjusting variable. Did they have access to the details of the values used for the calculation of the score. If so, it could have been interesting to perform another logistic regression model with all of these variables, in a sensitivity analysis for example.
Thank you for your comment. We believe that a score could be an appropriate measure of comorbidity due to the fact that it includes different items summarizing dysfunction of several organs. It is a relatively new way of stratifying risk populations in renal patients. This point has been added at the beginning of the conclusions paragraph.
Minor comments:
- typo on page 5, line 146: “The OR for IHM was 1.11” instead of “1.13”, according to table 5.
Thank you, done.
- Page 5, line 53. I am not sure that the abbreviation CCI has been defined in the text (I guess it stands for Charlson Comorbidity Index?)
Thank you, done.
- typo on page 5, line 174: “conducted study on more than cases in 53,000 dialysis…
Thank you, done.
